# Spin skyrmion gaps as signatures of strong-coupling insulators in magic-angle twisted bilayer graphene

Jiachen Yu [1,2,8], Benjamin A. Foutty[2,3,8], Yves H. Kwan[4,8], Mark E. Barber [1,2], Kenji Watanabe [5], Takashi Taniguchi [6], Zhi-Xun Shen [1,2,3,7], Siddharth A. Parameswaran [4] & Benjamin E. Feldman [2,3,7] ✉

The flat electronic bands in magic-angle twisted bilayer graphene (MATBG) host a variety of correlated insulating ground states, many of which are predicted to support charged excitations with topologically non-trivial spin and/or valley skyrmion textures. However, it has remained challenging to experimentally address their ground state order and excitations, both because some of the proposed states do not couple directly to experimental probes, and because they are highly sensitive to spatial inhomogeneities in real samples. Here, using a scanning single-electron transistor, we observe thermodynamic gaps at even integer moiré filling factors at low magnetic fields. We find evidence of a field-tuned crossover from charged spin skyrmions to bare particle-like excitations, suggesting that the underlying ground state belongs to the manifold of strong-coupling insulators. From the spatial dependence of these states and the chemical potential variation within the flat bands, we infer a link between the stability of the correlated ground states and local twist angle and strain. Our work advances the microscopic understanding of the correlated insulators in MATBG and their unconventional excitations.

Magic-angle twisted bilayer graphene (MATBG) has emerged as a remarkably rich platform to investigate correlated ground states of strongly interacting electrons[1–6]. The observation of insulating electrical transport when an integer number $v$ of electrons/holes occupy each moiré unit cell[1,3,6–10] was one of the earliest indications of strong electronic correlations in MATBG and remains among their most salient manifestations. However, the nature of these correlated insulators and the novel excitations they host is still a major puzzle under active investigation. In contrast to the abundance of transport signatures of insulating ground states in samples without hBN alignment, there is little evidence of truly gapped ground states in the few measurements

that directly probe the thermodynamic density of states[11–16] (the only reported example is[13]). Instead, such thermodynamic measurements consistently reveal a robust cascade of spin/valley flavor phase transitions which are present up to temperatures $T \approx 100$ K[13]. These transitions occur between gapless broken symmetry states that may be viewed as the parent correlated states from which the flavor-polarized insulating ground states emerge at low temperatures[17]. It is therefore imperative to corroborate the existence of true gapped ground states at commensurate filling from a thermodynamic perspective.

The spin, valley, and sublattice degrees of freedom in MATBG conspire with the narrow moiré bands to form a near-degenerate

[1]Department of Applied Physics, Stanford University, Stanford, CA 94305, USA. [2]Geballe Laboratory of Advanced Materials, Stanford, CA 94305, USA. [3]Department of Physics, Stanford University, Stanford, CA 94305, USA. [4]Rudolf Peierls Centre for Theoretical Physics, University of Oxford, Oxford OX1 3PU, UK. [5]Research Center for Electronic and Optical Materials, National Institute for Materials Science, 1-1 Namiki, Tsukuba 305-0044, Japan. [6]Research Center for Materials Nanoarchitectonics, National Institute for Materials Science, 1-1 Namiki, Tsukuba 305-0044, Japan. [7]Stanford Institute for Materials and Energy Sciences, SLAC National Accelerator Laboratory, Menlo Park, CA 94025, USA. [8]These authors contributed equally: Jiachen Yu, Benjamin A. Foutty, Yves H. Kwan. ✉e-mail: bef@stanford.edu

manifold of competing ground states[18–24]. In the absence of extrinsic factors such as strain or substrate alignment, theoretical considerations single out a specific "Kramers intervalley-coherent" (KIVC) insulator as the energetically favored candidate at moiré filling factors $\nu = 0$ and $\pm 2$[18,19,22]. However, the KIVC state is part of a larger family of "strong-coupling" insulators, driven by interaction strength that is much larger than bandwidth. These insulators exhibit similar topological structure: they are formed by a coherent superposition of flavor-polarized flat Chern bands, and their energetic hierarchy is sensitive to details of the modeling. Intriguing real-space patterns as well as unconventional collective excitations are predicted for these many-body ground states[17,18]. Furthermore, realistic departures from this idealized limit such as heterostrain between the two graphene sheets and local variations in twist angle can qualitatively alter the phase diagram, and have been theoretically shown to stabilize distinct but closely-competing alternatives to the strong-coupling states[23–26]. Establishing the precise nature of the correlated states that emerge from this delicate competition is thus a challenging experimental question, whose answer can vary even within a single sample.

Studying the spectrum of excitations can help identify the ground state, as it can carry an imprint of the underlying broken symmetry insulator. Specifically, it has been predicted[17,27–29] that the combination of strong correlations and nontrivial band topology that stabilizes a given strong-coupling insulator can also trigger a local deformation of the flavor degrees of freedom when charges are added or removed, so that the lowest-energy charged excitations are spin/pseudospin skyrmions rather than single electrons or holes. This could also have important implications for superconductivity in MATBG: it has been proposed that charge-$e$ pseudospin skyrmions of the KIVC state could pair into charge-$2e$ bosons that condense to drive the system into a superconducting state[17]. While there is some numerical support for this scenario[28], experimental evidence has remained elusive, in part due to the challenge of directly accessing the pseudospin degrees of freedom.

## Results

In this work, we use high-resolution local electronic compressibility measurements conducted with a scanning single-electron transistor (SET) to demonstrate the existence of thermodynamically gapped states at $\nu = \pm 2$. These states are present at zero magnetic field $B$ and their gap sizes are constant or increasing at low fields, but decrease linearly at moderate fields. Using Hartree-Fock calculations and symmetry analysis, we show that this nontrivial gap dependence can be rationalized in terms of a ground state with spin skyrmions as the lowest-lying charge excitations at small fields, yielding to conventional electrons and holes at larger $B$. The experimental observation of spin

skyrmions at $\nu = \pm 2$ is most naturally explained by a ground state whose pseudospin order corresponds to that of a strong-coupling insulator, among which the KIVC insulator is found to be the most energetically favorable candidate. However, since our experiment does not directly couple to the flavor degree of freedom, we cannot rule out the possibility of a related strong-coupling state which can also exhibit spin skyrmions. In the same regions where we observe a thermodynamic gap at $\nu = \pm 2$, we also find evidence for a thermodynamic gap at the charge neutrality point (CNP, $\nu = 0$) as well as a smaller total chemical potential change across the flat bands relative to areas in which the gaps are absent. These observations are consistent with a theoretical picture in which the strong-coupling state is destabilized as strain and/or twist angle variations broaden the flat bands and suppress the effect of interactions. Our work thus provides experimental evidence for spin skyrmions in MATBG, and gives microscopic insight into the role of spatial inhomogeneities in tuning the delicate balance between its competing ground states.

We first discuss the signatures of correlated insulating ground states at zero magnetic field. Figure 1a shows a line cut of the inverse compressibility $d\mu/dn$ as a function of spatial position within the sample. We observe a sawtooth-like pattern, which has been extensively reported in MATBG[11–16,30], throughout the measured region. Superimposed on this background, the data also reveal incompressible peaks at densities corresponding to $\nu = \pm 2$ in certain locations (Fig. 1b). These features represent thermodynamically gapped states, and the location at which each respective state is most robust is distinct. The incompressible peak at $\nu = 2$ is most pronounced at $\theta \approx 1.14°$, and is observed over 300 nm along the measured trajectory. Its gap size as a function of spatial position is shown in Fig. 1c (see Methods). We also indicate the corresponding $\theta$ in Fig. 1c, but note that strain, which is not directly probed in our experiment, may also play a role in the observed spatial dependence. A weaker incompressible state occurs at $\nu = −2$ at angles near $\theta = 1.18°$. The gaps are observed only within a relatively small area and were absent in other previously measured low-disorder areas within the sample[16]. This suggests that stabilizing these fragile states requires fine-tuning parameters such as twist angle and strain.

To clarify the nature of the correlated ground states, we investigate their dependence on the perpendicular magnetic field. Figure 2a–c shows $d\mu/dn$ as a function of $\nu$ and $B$, measured at locations with $\theta = 1.14°$ and $\theta = 1.18°$ (distinct from that in Fig. 1a, b), respectively. The gapped phase at $\nu = \pm 2$ survives up to $B \approx 6$ T and is not sloped in the $\nu$ - $B$ plane, indicating it has Chern number $C = 0$. The high-field Hofstadter (Chern) insulators in Fig. 2a (see Supplementary Information Section 3) follow the same universal pattern reported previously, independent of the existence of the $\nu = \pm 2$ insulators. The field dependence of the thermodynamic gaps $\Delta_\nu$ to the lowest available

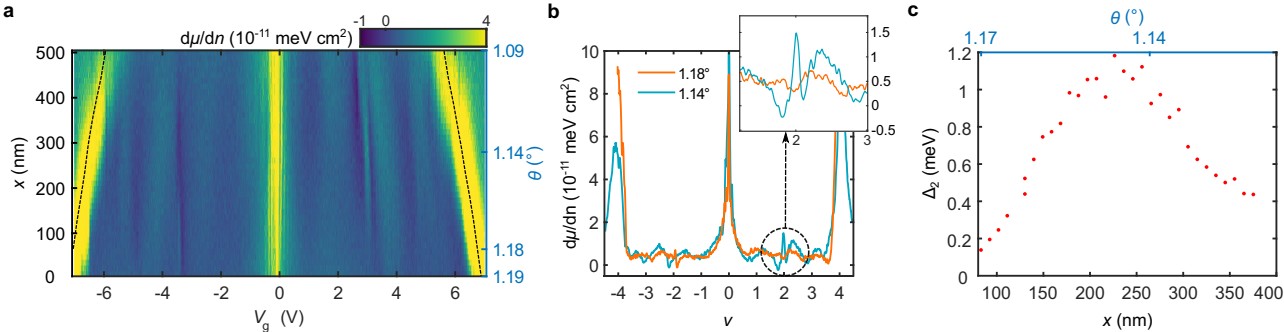

**Fig. 1 | Thermodynamically gapped states at zero magnetic field in magic-angle twisted bilayer graphene (MATBG). a** Inverse electronic compressibility $d\mu/dn$ as a function of gate voltage $V_g$ measured along a linear trajectory. Dashed lines denote gate voltages corresponding to filling factors $\nu = \pm 4$. **b** Individual line cuts of $d\mu/dn$ as a function of $\nu$ measured at fixed locations in (**a**) with respective twist

angles of $\theta = 1.14°$ (orange) and $\theta = 1.18°$ (blue). Inset, zoom in showing the incompressible peak near $\nu = 2$ at $\theta = 1.14°$. **c** Extracted thermodynamic gap $\Delta_2$ of the $\nu = 2$ correlated insulator as a function of position and twist angle along the spatial line cut in (**a**).

 

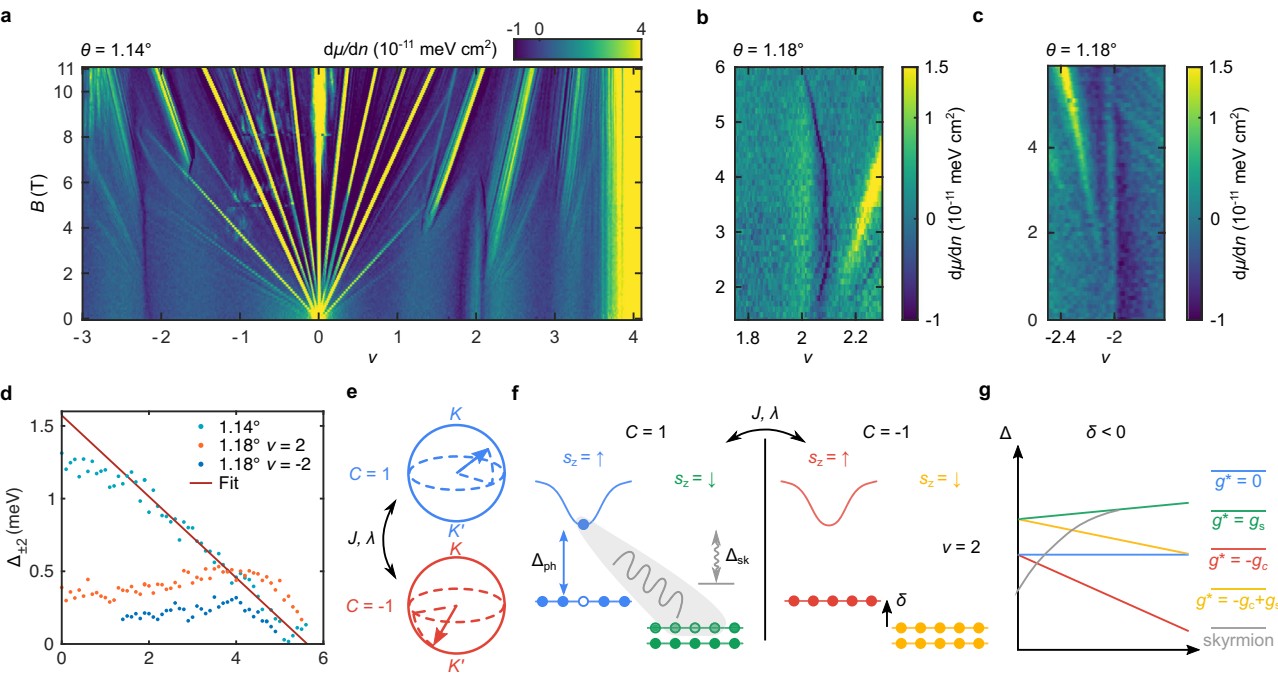

**Fig. 2 | Magnetic field dependence of the strong-coupling insulator.** **a** $d\mu/dn$ as a function of $\nu$ and perpendicular magnetic field $B$ at the $\theta = 1.14°$ location in Fig. 1a. Data is truncated beyond the limits of the colorbar. **b**, **c** $d\mu/dn$ measured at a $\theta = 1.18°$ location (distinct from that in Fig. 1a, b). **d** Thermodynamic gap $\Delta_{\pm 2}$ at $\nu = \pm 2$ as a function of $B$ at the respective spatial locations in (**a**–**c**). A linear fit of the $\theta = 1.14°$ gap (red line) for $B \gtrsim 1.8$ T. **e** Schematic of $\nu = 2$ strong-coupling bands symbolized by two pseudo-spinors in the valley Bloch spheres of each Chern sector. Dispersion and deviations from the chiral ratio manifest as couplings $J, \lambda$ which select the strong-coupling ground state. Note that the spin structure has not been indicated. **f** Schematic mean-field band diagram of a strong-coupling insulator ground state, assuming for simplicity a spin ferromagnetic configuration. The spin up/down ($\uparrow$/$\downarrow$) bands in the $C = +1/-1$ sectors are colored blue, green, red, and yellow, respectively. $\Delta_{ph}(\Delta_{sk})$ denotes particle-hole (skyrmion) gap. Other types of particle-hole gap involving different pairs of bands are not indicated. **g** Energy gaps as a function of $B$ for different possible charged excitations at $\nu = 2$ corresponding to different spin and orbital Zeeman couplings. The gap $\Delta_2$ is set by the lowest energy excitation. The slope of each particle-hole gap (solid lines) is given by the effective $g^*$ from the relation: $E(B) = E(B=0) + g^* \mu_B B$, and depends on the direction of spin/orbital moments relative to the perpendicular magnetic field. We assume $g_c > g_s \geq 2$ ($g_c$, orbital $g$-factor of the Chern bands; $g_s$, spin $g$-factor; $\mu_B$, Bohr magneton). A $B = 0$ energy offset $\delta < 0$ (see Supplementary Fig. S3 for the case of $\delta > 0$, and Supplementary Information Section 4) controls the relative band alignment at low fields. The field-dependence of the skyrmion gap (dashed lines) is governed by the Zeeman suppression of spin flips.

charged excitations at filling factor $\nu$ is plotted in Fig. 2d (see Methods and Supplementary Fig. S2). In the $\theta = 1.14°$ location, $\Delta_2$ decreases linearly when $B \gtrsim 1.8$ T, with a slope corresponding to an effective $g$-factor $g^* = 4.85 \pm 0.03$ (Fig. 2c, the error primarily comes from uncertainty in determining the lower bound of the linear fit). However, the magnitude of the gap at low fields is smaller than predicted by extrapolating this linear trajectory. Even more strikingly, in the $\theta = 1.18°$ location, $\Delta_{\pm 2}$ slightly increases with $B$ up to about 4–4.5 T, and then decreases linearly with a similar $g^*$ as in the 1.14° location (Fig. 2d). Qualitatively similar gap dependence is also observed at $\nu = 2$, suggesting similar excitations are realized (see Supplementary Information Section 2). The large $g^*$ extracted suggests the presence of an electron orbital moment that couples to the Zeeman field, even in the absence of substrate-induced $C_{2z}$ symmetry breaking and without any zero-field quantum anomalous Hall signatures at odd integer fillings. This orbital moment has its origin in the Berry curvature carried by the underlying bands[31,32], and contributes to an additional orbital $g$-factor $g_c$: $g^* = g_s + g_c$ ($g_s = 2$ for spin Zeeman coupling).

The gap evolution at $\nu = \pm 2$ presented above can be understood within the framework of strong-coupling states. In the absence of strain, strong-coupling theory predicts[18–24] that the ground state at even integer fillings is part of a family of correlated insulators, whose topological structure at $\nu = \pm 2$ is reminiscent of a quantum Hall ferromagnet consisting of two filled Landau levels with opposite $C$ (Fig. 2e, f). In the following, we assume spin ferromagnetism appropriate for a ferromagnetic Hund's coupling for simplicity, but our conclusions are similar for the spin-valley-locked state favored by an

antiferromagnetic coupling (see Supplementary Information Section 5). We find that the gap at $B = 0$ is controlled by spin skyrmions (Fig. 2e–g, Supplementary Fig. S3) that dominantly involve a single Chern sector (see Supplementary Information Section 4) and are the lowest-energy charged excitations at low fields. The dominant effect of an applied $B$ field is to impose a large Zeeman penalty and reduce the skyrmion size, which also raises its Coulomb energy cost. Hence the gap initially increases at small fields, until the skyrmion excitations become energetically more costly than a single particle-hole pair. This occurs when the dashed skyrmion line in Fig. 2g crosses the lowest available particle-hole excitation, whose nature depends on quantitative details such as the energy difference $\delta$ between filled bands of the renormalized band structure. Above this critical field, the thermodynamic gap is controlled by the lowest-energy particle-hole excitation, which closes linearly because of the coupling to the orbital magnetization of the Chern bands (Fig. 2g, Supplementary Fig. S3).

To confirm the above picture, we have performed Hartree-Fock calculations on the Bistritzer-MacDonald model that demonstrate the stability of spin skyrmions in MATBG and their fingerprint on the non-monotonic gap (see Supplementary Information Sections 4 and 6, which also includes a discussion of alternative mechanisms). Note that within our modeling, the ground state at even integer filling is the KIVC state, but we expect similar behavior from other strong-coupling candidates. We find that the lowest-energy hole excitation at $\nu = 2$ is a spatially localized skyrmion in one Chern sector with a topologically non-trivial spin texture (Fig. 3a, b) and a non-topological pseudospin

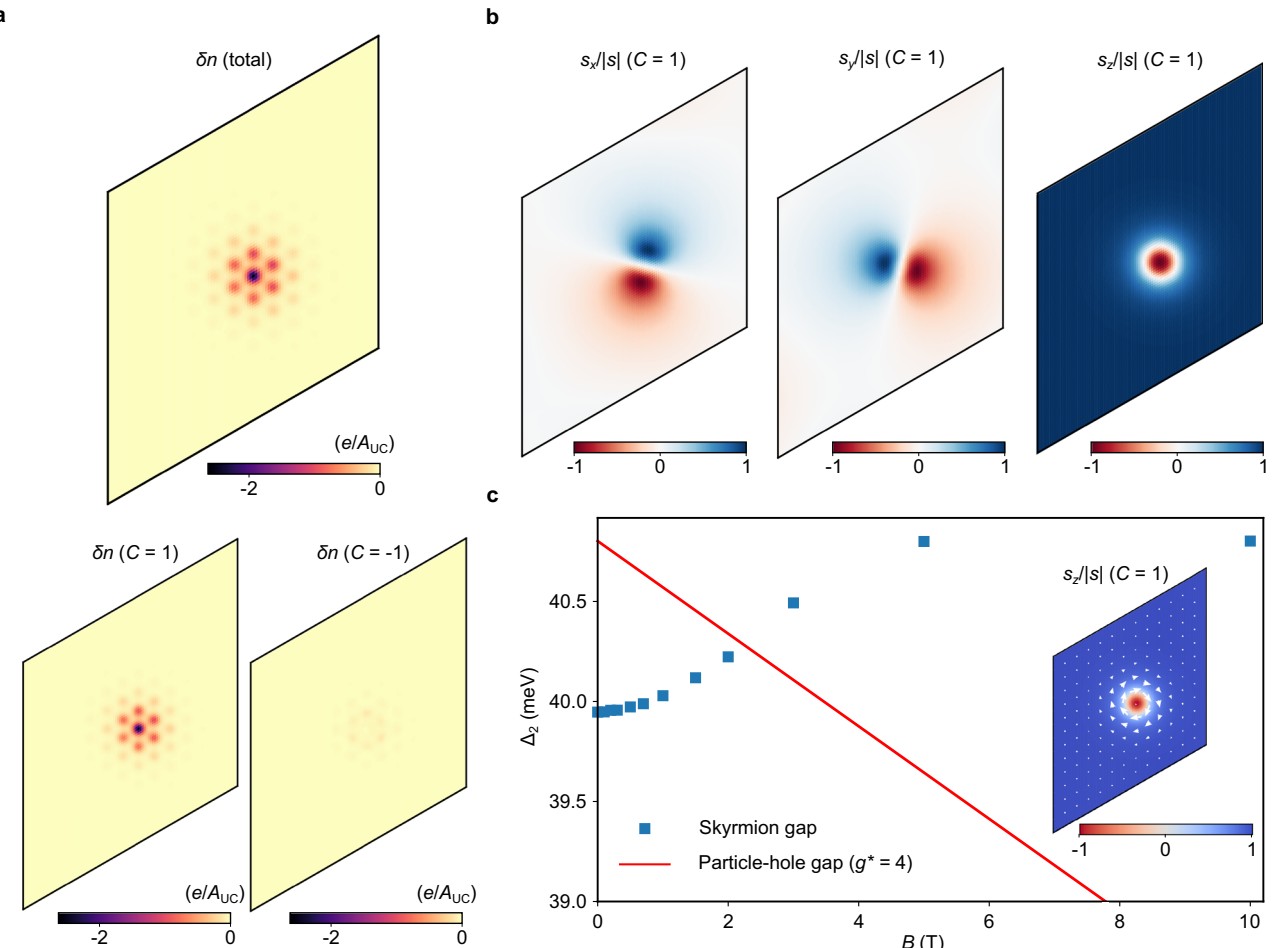

**Fig. 3 | Hartree-Fock calculations of spin skyrmions of the strong-coupling state at $v = 2$. a** Real-space charge density of a hole skyrmion, measured relative to the insulator, in the periodic simulation area of $13 \times 13$ moiré unit cells. The parent insulator in Hartree-Fock is the KIVC state. The charge modulation is localized in one Chern sector (here we show it in $C = 1$; a degenerate excitation can be found in the opposite Chern sector by time-reversal symmetry). **b** Real-space spin structure of the skyrmion in the $C = 1$ sector. **c** $\Delta_2$ as a function of $B$, reflecting two possible excitations. Blue squares are numerically computed values of the skyrmion gap in the presence of the spin Zeeman effect. Red line indicates evolution of the particle-hole gap from zero field assuming an additional effective orbital coupling $g^* = 4$. This leads to a crossover from the skyrmion gap at small fields to the particle-hole gap at a critical field $B \approx 5$ T. The Hartree-Fock calculations here are expected to overestimate the gap. $w_{AA} = 30$ meV, $\varepsilon_r = 6$. Inset, spin projection of the skyrmion along the Zeeman axis in the $C = 1$ sector. In-plane projection is denoted with arrows.

modulation (see Supplementary Information Section 4). Skyrmions do not form on electron-doping due to the larger bandwidth of the bands above the chemical potential[28,33]. The initial increase of $\Delta_2$ upon including spin Zeeman coupling at low fields is consistent with the discussion above (blue squares in Fig. 3c), and the gap closes at higher fields due to the orbital coupling to the magnetic field (red line). While precise details of the microscopic modeling and fluctuations beyond mean-field will quantitatively affect the results, our calculations provide a proof-of-principle demonstration that spin skyrmions remain the lowest-energy hole excitations of the $v = 2$ strong-coupling state even in realistic settings where the flat bands deviate significantly from idealized Landau level-like topological structure.

In the presence of strain, the only gapped state energetically competitive with the strong-coupling insulators is the "incommensurate Kekulé spiral" (IKS) state[23]. This is a spin-singlet state at $v = \pm 2$ whose gap strictly decreases with increasing $B$, and is hence difficult to reconcile with the observed field dependence of the gap at $\theta = 1.18°$ (see Supplementary Information Section 4). The observed CNP gap (see below), which is absent for strains large enough to stabilize the IKS state, is further evidence against this possibility for both the $\theta = 1.14°$ and $1.18°$ regions.

Under the low-strain conditions for which strong-coupling order is stabilized at $v = \pm 2$, theory also predicts strong-coupling order at $v = 0$ and hence an interaction-driven charge gap is expected at the CNP[18–20,22–24], even in the absence of substrate alignment. So far, only one transport experiment[6] has reported an activation gap at the CNP in non hBN-aligned devices. This could be due to the sensitive dependence of the low-energy physics on fine-tuning parameters like twist angle, local disorder, and strain, which complicates the interpretation of global measurements. In contrast, the scanning SET can measure locally in widely separated areas, which allows us to probe and compare multiple locations whose microscopic parameter configurations differ substantially. We find that the slope of $\mu(v)$ in the vicinity of CNP is significantly larger in the areas where $v = \pm 2$ gaps are observed (denoted as Area 1) compared to a far-separated region of the same device in which no gaps are present at $B = 0$ (denoted as Area 2 and characterized in detail in ref. 16), suggestive of a spontaneous gap at the CNP (Supplementary Information Section 7). In Fig. 4a, we compare $\mu(v)$ at the $\theta = 1.14°$ location (Area 1) in Fig. 1b and the $\theta = 1.06°$ location (Area 2). The shape of $\mu(v)$ near the CNP in Area 2 is consistent with that expected from a massless Dirac dispersion on the electron side[16], whereas in Area 1, it changes significantly more rapidly (Fig. 4a,

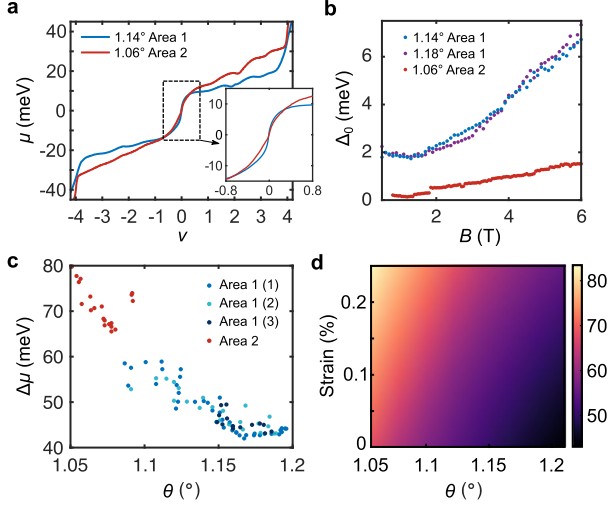

**Fig. 4 | Signature of spontaneous gap at charge neutrality and spatial dependence. a** Chemical potential $\mu$ at $B = 0$ as a function of $\nu$ at the $\theta = 1.14°$ location and the $\theta = 1.06°$ location reported in ref. 16. Inset, zoom in of the dispersion near $\nu = 0$. **b** Thermodynamic gap $\Delta_0$ at charge neutrality as a function of $B$ in the same locations from (**a**) as well as that from Fig. 2b. **c** Change in chemical potential $\Delta\mu$ from $\nu = -4$ to 4, extracted from four distinct spatial line cuts taken in Area 1 and Area 2. **d** Numerical calculation of $\Delta\mu$ as a function of twist angle and strain for an example set of parameters $w_{AA} = 60$ meV, $\varepsilon_r = 6$, showing that $\Delta\mu$ rises with decreasing $\theta$ and increasing strain.

inset), despite the significantly smaller total change in chemical potential across the flat bands, $\Delta\mu_{tot} = \mu(\nu = 4) - \mu(\nu = -4)$.

While it is experimentally challenging to distinguish a small thermodynamic gap from a vanishing density of states at the Dirac point at $B = 0$, the evolution with magnetic field can provide additional evidence to help differentiate them. Figure 4b shows the extracted $\nu = 0$ zLL gap $\Delta_0$ as a function of $B$. In Area 2, $\Delta_0$ follows a linear trajectory that extrapolates to 0 at zero field, whereas in Area 1, it saturates at approximately 2 meV below about 2 T, exhibiting close quantitative agreement in the 1.14° and 1.18° locations (Fig. 4b). We note that the magnitude of $\Delta_0$ in nonzero $B$ is also much larger for Area 1, whereas the other zLL gaps show a similar magnitude (Supplementary Fig. S9) and are consistent with a phenomenological moiré valley splitting model[16]. Taken together, the data strongly suggest that the CNP in Area 1 is gapped (See Supplementary Section 7 for more detail, including microwave impedance microscopy measurements). Previously reported single-particle gaps induced by hBN alignment in MATBG were 7–10 meV[30,32], much larger than the extrapolated gap size in this work, suggesting that the CNP gap may form spontaneously as a result of interactions rather than substrate alignment. This is consistent with theoretical predictions as noted above.

The mechanism that gives rise to two distinct types of behaviors in the same device is unknown; we speculate that the ground states are highly sensitive to twist angle and strain[23–26,34], and hence postulate a minimal explanation in terms of local heterostrain variations. Whereas for small strain the strong-coupling insulator is the energetically favored ground state at $\nu = 0$ and $\pm 2$, even modest heterostrains are known to degrade the $\nu = 0$ gap in favor of a gapless semimetal[24–26], and to strongly suppress the strong-coupling gap at $\nu = \pm 2$. This interpretation is also consistent with the $\Delta\mu_{tot}$ of the low energy bands, which is much smaller in Area 1 than Area 2 (Fig. 4c). Our calculations generally predict a trend of decreasing $\Delta\mu_{tot}$ with decreasing strain and with increasing twist angle (Fig. 4d), with quantitative values dependent on microscopic parameters. However, the large magnitude of the difference in $\Delta\mu_{tot}$ between Area 1 and Area 2 suggests that twist angle alone cannot explain the difference and strain is lower in Area 1

(Supplementary Information Section 8). Although a gap is expected to eventually open as the IKS state emerges for sufficiently large strains, mean-field theory predicts a steep suppression of the strong-coupling gap before this occurs. Our data are consistent with a scenario in which Area 2 has an intermediate level of strain, such that a gap is absent or below the resolution of our measurement.

In conclusion, our measurements provide evidence of thermodynamically gapped ground states at $\nu = 0, \pm 2$. The magnetic field dependence of the $\nu = \pm 2$ gapped state is consistent with a strong-coupling insulator whose low field charged excitations are spin skyrmions. Spatially resolved imaging shows that these states are stabilized when the change in chemical potential across the flat bands is small, indicating relatively low strain. In a broader context, our work establishes measurement of thermodynamic gaps as a powerful diagnostic tool for probing unconventional charged excitations. These results further motivate direct imaging of flavor ordered ground states and their topological excitations[35–37], especially efforts to experimentally determine whether certain types of strong-coupling ground states support pseudospin skyrmion excitations, and their relation to superconductivity.

## Methods

### Sample fabrication

The MATBG device is fabricated using standard dry-transfer techniques, followed by standard lithographic patterning, as detailed in ref. 16. During the device fabrication process, we have avoided hBN substrate alignment to the graphene layers, which has been confirmed with measurements presented above and also in ref. 16 at an independent location in the same device.

### SET measurement

An SET sensor with a diameter of 50–80 nm was brought to < 50 nm above the MATBG sample surface. The resulting spatial resolution is about 100 nm. We modulate the MATBG and gate voltages, $V_{2d,ac}$ and $V_{g,ac}$, respectively, and measure inverse compressibility $d\mu/dn \propto I_{g,ac}/I_{2d,ac}$, where the corresponding currents $I_{g,ac}$, $I_{2d,ac}$ are demodulated from the current through the SET probe at their respective modulation frequencies using standard lock-in techniques. A d.c. offset voltage $V_{2d,dc}$ is further applied to the sample to maintain maximum sensitivity of the SET and minimize tip-induced doping. All data presented are taken at 330 mK in an Unisoku USM1300 system with a customized microscope head.

### Gate capacitance and twist angle determination

The capacitance between gate and sample (*i.e.*, the conversion between density and applied gate voltage) was calibrated by measuring the slope of the Landau levels emanating from charge neutrality, which yielded a value consistent with geometric considerations. The twist angle is then calculated using $\theta(r) = \alpha\sqrt{\frac{\sqrt{3}n_s(r)}{8}}$ where $a = 0.246$ nm is graphene's lattice constant, and $n_s(r)$ is the carrier density at full filling.

### Extraction of $\Delta_2$

To extract the thermodynamic gaps, we identify either the points surrounding an incompressible peak at which $d\mu/dn$ crosses zero, i.e. the local extremum of $\mu(n)$, or the local minimum of $d\mu/dn$ if it never crosses zero. We numerically integrate $d\mu/dn$ between the filling factors corresponding to these two densities, and the resulting step in chemical potential $\Delta\mu$ is taken to be the gap size. However, if $d\mu/dn$ exhibits a minimum both sides of the incompressible peak, which arises if there is a large negative background from the sawtooth, then prior to integration, we shift the entire $d\mu/dn$ curve so that the less negative local minimum in $d\mu/dn$ is zero. This procedure will slightly change (< 30 μeV) the bare value of the extracted gap but has negligible effect on the extraction of $g$. An example

demonstrating the gap extraction procedure is shown in Supplementary Fig. S2.

## Hartree-Fock calculations

The properties and energetics of skyrmions of the strong coupling insulator at $v = 2$ were theoretically investigated using translation symmetry-breaking Hartree-Fock calculations of the Bistritzer-MacDonald model on a periodic system of size $13 \times 13$. In agreement with prior theoretical work, the numerical ground state at $v = 2$ is identified with the KIVC insulator. Upon hole-doping, the spin/pseudospin structure of the numerically obtained skyrmions is consistent with a non-linear sigma model analysis that captures the main symmetry-breaking terms of the Hamiltonian. The dependence of the thermodynamic gap $\Delta_2$ on the external Zeeman field was estimated by calculating the minimum energy Hartree-Fock solutions (including skyrmions) for total electron numbers $N_{v=2}$, $N_{v=2} \pm 1$, where $N_{v=2}$ corresponds to filling factor $v = 2$. The chemical potential range $\Delta\mu_{tot} = \mu(v = 4) - \mu(v = -4)$ was computed by evaluating the total energy for a band insulator with electron numbers $N_{v=-4}$ and $N_{v=4}$.

## Microwave impedance microscopy measurements

Microwave impedance microscopy (MIM) measurements were performed in a 3He cryostat with a custom scanner incorporating Attocube nanopositioner. An etched tungsten wire was used as the MIM probe and was attached to a quartz tuning fork for topographic sensing. During measurement the tip was held approximately 20 nm above the sample's surface. The measurements reported here were carried out at 6.8 GHz. At GHz frequencies the tip and sample are strongly capacitively coupled, enabling sub-surface sensing without requiring additional electrical contacts on the sample. MIM operates in the near-field limit and the spatial resolution is dictated by the tip diameter, ~100 nm, rather than the microwave wavelength.

## Data availability

The data that support the findings of this study are available from the corresponding authors upon reasonable request.

## Code availability

The codes that support the findings of this study are available from the corresponding authors upon reasonable request.

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

## Acknowledgements

We thank Erez Berg, Nick Bultinck, and Pablo Jarillo-Herrero for helpful discussions. Y.H.K. and S.A.P. thank Glenn Wagner, Nick Bultinck, and Steven H. Simon for collaboration on related theoretical work. This work was supported by the QSQM, an Energy Frontier Research Center funded by the US Department of Energy (DOE), Office of Science, Basic Energy Sciences (BES), under award no. DE-SC0021238. B.E.F. acknowledges a Stanford University Terman Fellowship and an Alfred P. Sloan Foundation Fellowship. Y.H.K. and S.A.P. acknowledge support from the European Research Council under the European Union Horizon 2020 Research and Innovation Programme, Grant Agreement No. 804213-TMCS. K.W. and T.T. acknowledge support from the JSPS KAKENHI (Grant Numbers 20H00354 and 23H02052) and World Premier International Research Center Initiative (WPI), MEXT, Japan. B.A.F. acknowledges a Stanford Graduate Fellowship. M.E.B. acknowledges support from the Marvin Chodorow Postdoctoral Fellowship of the Applied Physics Department, Stanford University. Part of this work was performed at the Stanford Nano Shared Facilities (SNSF), supported by the National Science Foundation under award ECCS-2026822.

## Author contributions

J.Y., B.A.F. and Y.H.K. contributed equally to this work. J.Y., B.A.F., and B.E.F. designed and conducted the scanning SET experiments. B.A.F. fabricated the sample. Y.H.K. and S.A.P. conducted the theoretical calculations. M.E.B and Z.-X.S. designed and conducted the MIM experiments. K.W. and T.T. provided hBN crystals. All authors participated in discussions and in writing of the manuscript.

## Competing interests

Z.-X.S. is a co-founder of PrimeNano Inc., which licensed the MIM technology from Stanford University for commercial instruments. The remaining authors declare no competing interests.
