## [Peer Review File · Nature Communications]

Reviewers' Comments:

Reviewer #1:

Remarks to the Author:

While I agree with referee 1 that there are still many unknowns associated with the findings, the authors have carefully considered all possible known scenario and discussed the limitations of their data. The revised manuscript is also further improved and will be a valuable addition to the field. Therefore, I strongly recommend publication in Nature Communications.

Reviewer #2:

Remarks to the Author:

The authors have addressed most of my concerns. I recommend the manuscript to be published.

Reviewer #3:

Remarks to the Author:

The authors report incompressibility measurements on a sample of twisted bilayer graphene using a scanning electron transistor (SET). This technique has the advantage of locally probing thermodynamic gaps in contrast to transport measurements which average over large areas of the sample. Given the sensitivity of the correlated phases to local variations in strain, disorder, and twist angle, these measurements provide valuable information on the possible correlated phases at different locations in the sample. The authors identify a large region in the sample where the cascade features previously observed close to integer filling are also accompanied by incompressibility peaks at $\nu=+2$, -2 , and at charge neutrality. The observation of a charge neutrality peak in a sample which is presumable unaligned with the hBN substrate is interpreted as an indication that the region studied has very small heterostrain. Despite the absence of local strain characterization in their experiment, their finding and interpretation is consistent with recent STM data from Yazdani's group which observed a charge neutrality gap in ultra-low strain regions of the sample and a semimetallic behavior in strained regions. This interpretation is also consistent with theoretical calculation which predicts an insulating strong coupling state at zero strain but a semi metal for intermediate values of heterostrain. Using this information, the authors infer that the $\nu=+2, -2$ are also strong coupling insulating states (spin polarized analogs of the state at charge neutrality) rather than the intervalley Kekule spiral (IKS) predicted for large heterostrain. The most striking finding of the manuscript is a non-monotonic dependence of the gap on magnetic field. This dependence is inconsistent with a spin-singlet IKS state. It is also inconsistent with single-particle excitations on top of the strong coupling states. The authors argue that such dependence is consistent with skyrmions being the lowest energy charge excitation and support their argument with numerical Hartree-Fock calculation for the skyrmion energies on hole doping the $\nu=+2$ state. In their response to previous criticism from the referees, the authors carefully examine and exclude all other possible sources for such dependence on field. Overall, the experimental data and theoretical analysis presents an intriguing albeit indirect evidence for non-trivial charge excitations in the system. In addition, this is the first measurement of a thermodynamic gap at charge neutrality in an unaligned sample. Therefore, I recommend the paper for publication but would like the authors to address one issue. The gaps predicted from Hartree-Fock for the insulating states are significantly larger than the gapped experimentally measured. This is a known issue and can be explained by local disorder or some other effect. However, given this issue, can the authors justify why the dependence of the gap on magnetic field obtained from the same calculation should be trusted? Isn't it possible that the unknown effect the reduces the gap also leads to more complicated dependence on the field?

Author response to reviewer comments - NCOMMS-23-22974-T

We thank all the reviewers for reading our revised manuscript and for the recommendation of its publication. Below we address remaining issues raised by the reviewers.

Reviewer #1

While I agree with referee 1 that there are still many unknowns associated with the findings, the authors have carefully considered all possible known scenario and discussed the limitations of their data. The revised manuscript is also further improved and will be a valuable addition to the field. Therefore, I strongly recommend publication in Nature Communications.

We thank the reviewer for providing valuable feedback during the review process and for recommending publication.

Reviewer #2

The authors have addressed most of my concerns. I recommend the manuscript to be published.

We thank the reviewer for providing valuable feedback during the review process and for recommending publication.

Reviewer #3:

The authors report incompressibility measurements on a sample of twisted bilayer graphene using a scanning electron transistor (SET). This technique has the advantage of locally probing thermodynamic gaps in contrast to transport measurements which average over large areas of the sample. Given the sensitivity of the correlated phases to local variations in strain, disorder, and twist angle, these measurements provide valuable information on the possible correlated phases at different locations in the sample. The authors identify a large region in the sample where the cascade features previously observed close to integer filling are also accompanied by incompressibility peaks at $\nu=+2$, -2 , and at charge neutrality. The observation of a charge neutrality peak in a sample which is presumably unaligned with the hBN substrate is interpreted as an indication that the region studied has very small heterostrain. Despite the absence of local strain characterization in their experiment, their finding and interpretation is consistent with recent STM data from Yazdani's group which observed a charge neutrality gap in ultra-low strain regions of the sample and a semimetallic behavior in strained regions. This interpretation is also consistent with theoretical calculation which predicts an insulating strong coupling state at zero strain but a semi metal for intermediate values of heterostrain. Using this information, the authors infer that the $\nu=+2, -2$ are also strong coupling insulating states (spin polarized analogs of the state at charge neutrality) rather than the intervalley Kekule spiral (IKS) predicted for large heterostrain. The most striking finding of the manuscript is a non-monotonic dependence of the gap on magnetic field. This dependence is inconsistent with a spin-singlet IKS state. It is also

inconsistent with single-particle excitations on top of the strong coupling states. The authors argue that such dependence is consistent with skyrmions being the lowest energy charge excitation and support their argument with numerical Hartree-Fock calculation for the skyrmion energies on hole doping the $\nu=+2$ state. In their response to previous criticism from the referees, the authors carefully examine and exclude all other possible sources for such dependence on field. Overall, the experimental data and theoretical analysis presents an intriguing albeit indirect evidence for non-trivial charge excitations in the system. In addition, this is the first measurement of a thermodynamic gap at charge neutrality in an unaligned sample. Therefore, I recommend the paper for publication but would like the authors to address one issue. The gaps predicted from Hartree-Fock for the insulating states are significantly larger than the gapped experimentally measured. This is a known issue and can be explained by local disorder or some other effect. However, given this issue, can the authors justify why the dependence of the gap on magnetic field obtained from the same calculation should be trusted? Isn't it possible that the unknown effect that reduces the gap also leads to more complicated dependence on the field?

We thank the reviewer for recommending publication, and for the additional question to address in more detail prior to publication. Below, we address the remaining question regarding the gap dependence.

As mentioned by the reviewer, Hartree-Fock typically overestimates the magnitude of the gap, with one possible reason being disorder of some sort. We do not believe that charge impurities play a significant role, given the cleanliness of graphene samples and the fact that the compressibility data evolve smoothly with position. This leaves long-wavelength sources of disorder such as twist-angle and strain modulations. To address this point, we emphasize that another common way to experimentally measure a gap in TBG is by fitting transport to an activated behaviour, which is more susceptible to disorder-driven gap suppression and may be plagued by attendant complications. This is because the transport "gap" is not really a thermodynamic quantity and depends on various other parameters in a much more complex way than the thermodynamic gap. In contrast, the measurement here is thermodynamic. Furthermore the data presented shows that we see multiple behaviours in one sample. The most plausible and simplest explanation of this is that each region studied in compressibility is locally relatively disorder-free, giving more reason to trust the calculations.

We believe that a likely effect that suppresses the gap is fluctuations beyond mean-field, and we would (without any good reason to the contrary) expect this to be the dominant mechanism as well as monotonic in B . Intuitively, we anticipate the magnitude of fluctuations to depend roughly on the size of the gap itself. While this could alter the scale of the non-monotonicity, we find it unlikely that it could itself seed non-monotonicity. On the other hand, the Zeeman suppression of the spin skyrmion and its competition with single-particle excitations is a transparent physical principle that we expect to be robust beyond the mean-field regime. We also

note qualitative agreement in the skyrmion gap between the Hartree-Fock predictions and experimental measurements in *Phys. Rev. Lett.* **75**, 4290 (1995), though for a rather different system. We therefore conclude that these unknown effects should not affect the validity of the theoretical approach adopted in the main text nor the qualitative robustness of the results. We have incorporated the above discussion into the Supplementary Information.

Reviewers' Comments:

Reviewer #3:

Remarks to the Author:

The authors have addressed all my concerns. I recommend for publication.

Author response to reviewer comments

Reviewer #3 (Remarks to the Author):

The authors have addressed all my concerns. I recommend for publication.

Our response:

We thank the reviewer for their positive recommendation for publication.